# Gender Bias in the Australian Construction Industry: Women's Experience in Trades and Semi-Skilled Roles

Sarah Holdsworth *, Michelle Turner and Orana Sandri

School of Property Construction and Project Management, RMIT University, Melbourne, VIC 3001, Australia; michelle.turner@rmit.edu.au (M.T.); orana.sandri@rmit.edu.au (O.S.)
* Correspondence: sarah.holdsworth@rmit.edu.au

**Abstract:** While most industry sectors in the Australian workforce have consistently improved regarding the participation of women, the construction industry remains an exception. Despite multiple gender equality initiatives and regulations at all levels of the Australian Government, the proportion of women employed in the construction industry has steadily declined. In 2020, only 1% of the trades and technician positions in the Australian construction industry were filled by women. In this qualitative study, interviews were undertaken with 43 women working in trades and semi-skilled roles to identify the varying types of gender biases experienced by women and the resultant harms that these biases create. Biases consisted of challenges to credibility; characteristics of the work environment comprising support, amenities, conditions of employment, career development, and access to meaningful work; gender stereotypes about women's work roles; and objectification. Each of these biases has a cumulative impact on women, leading to systemic and structural discrimination. The implications and suggestions for strategies to address biases are discussed, including the need for structural interventions to create epistemic justice and recognition for women working in construction.

**Keywords:** women; trades; epistemic injustice; construction

## 1. Introduction

In Australia, the Workplace Gender Equality Agency (2022) reports that the workforce participation rate is 62.1% for women and 70.4% for men. While most industry sectors in the Australian workforce have consistently improved regarding the participation of women, the construction industry remains an exception. There have been longstanding and persistent problems with gender equality in the Australian construction industry, where women are significantly under-represented across all roles and sectors, and previous attempts to address this inequality have largely failed (Jones et al. 2017).

Despite multiple gender equality initiatives and regulations at all levels of the Australian Government over the last 35 years, the proportion of women employed in the construction industry has steadily declined. In 2006, women occupied 17% of the entire construction workforce (Australian Bureau of Statistics 2006), dropping to 12% in 2016 (Australian Bureau of Statistics 2016), and decreasing further to 11% in 2020 (Master Builders Association 2020). The proportion of women employed in trade occupations is even lower; only 1% of the trades and technician positions in the construction industry were filled by women in 2020 (Master Builders Association 2020). Large companies with over 100 employees employ slightly more women, with women representing 3.1% of their trade workforce (Workplace Gender Equality Agency 2020). The recruitment of women into construction-related vocational training is also of concern, as male student enrolments and completions consistently dominate at almost 99 per cent (Australian Bureau of Statistics 2016). At each career stage—recruitment, retention and progression—construction is overrepresented by men (Galea 2018).

Women choose not to enter the construction industry due to its poor reputation as a workplace that presents cultural, structural, and action-oriented barriers undermining

their participation, retention and progression (Dainty et al. 2001). The deeply entrenched masculine culture in construction has shaped, sustained, and fostered the highly gendered work structures, processes, practices, and norms, which disadvantage and exclude women from developing a career. The industry's culture presents multiple stressors through its macho, sexist, and conflict-ridden culture, the exclusive boys' club mentality, and the lack of female networks and role models. Women are regularly "locked out" of the construction industry due to the dominant male culture and stereotyped gendered roles that view women as not capable or suited to the work. Women working within the industry are often prevented from access to job-related education and participation in meaningful work that aligns with their skills, which limits secure and supported career pathways (Dainty et al. 2001; Galea et al. 2015; George and Loosemore 2019).

Gender equity reflects a workplace where everyone is treated fairly and equally and granted fair opportunities regardless of gender, ensuring equal access to rewards, resources, and opportunities (Workplace Gender Equality Agency 2016). Galea (2018) argues that gender equality is grounded in "arguments of social justice, equality and fairness" (p. 5) and that discrimination is inexcusable. Further, gender equality presents a business case given it "reduces staff turnover and attrition, widens the talent pool of candidates, enhances the talent attraction of companies, addresses skills shortages and develops an adaptive and innovative workforce" (p. 5). Gender equality in the workforce leads to higher productivity and economic growth (Workplace Gender Equality Agency 2016). Further, improved gender equality could counter the industry's poor public image and increase the attraction of women considering a career in construction (Galea 2018).

The success of future initiatives to address and improve gender inequality in the construction industry requires an understanding of women's experiences and the factors that inform their career choices. Most research on women in the construction industry has focused on women in professional roles or not in an Australian context (Rosa et al. 2017). Consequently, there is a significant lack of understanding and a pressing need for more research into women's experiences in trades and semi-skilled roles (Navarro-Astor et al. 2017; Ness and Green 2012). Existing research indicates that women's experiences in construction trades and semi-skilled roles are shaped by the nature of the work and by intransigent gender discrimination and inappropriate conduct in the workplace. There is a need, therefore, for an in-depth understanding of the experiences of women in trades so as to address organisational injustices around gender inequality and harassment (Chan et al. 2020).

This research responds to calls from Ness and Green (2012) for the need for researchers to "immerse themselves in the intricacies of construction and how the labour market operates" (Ness and Green 2012) and "embrace a knowledge of the construction process and the technologies and skills which are routinely mobilised—and by whom" (Ness and Green 2012). This paper is informed by research using semi-structured interviews with 43 women to understand the challenges women working in trades and semi-skilled construction roles face in the Australian construction industry. The research questions framing this study consist of:

1. What biases do women working in construction experience?
2. What harm do women experience as a result of gender-based bias?
3. What response is required by the industry to eliminate or control gender-based bias and associated harm?

Reporting on the findings from these interviews, this paper highlights the varying types of gender biases experienced by women and the harm these create. The implications and suggestions for strategies for addressing such biases are then discussed.

## 2. Gendered Values Shaping Construction

The belief that construction is exclusively for men results from an accepted "reality" that masculinity and femininity are perceived and aligned to different qualities, with masculinity aligned to positive attributes such as independence and rationality and femininity with weaker characteristics such as dependence, emotion, and empathy (Galea 2018). This

duality then informs gendered values determining industry-appropriate, acceptable masculine and feminine forms of behaviour and a gendered logic determining who should and should not work in the sector (Chappell and Waylen 2013; Galea 2018). This socially constructed gendered logic subsequently informs rules and operating systems that determine appropriate actions and form institutions (systems of work) within which practices are acceptable. Resulting from this are socially accepted rules about "men's work" and "women's work" that allocate and separate specific roles, actions, and traits based on gender (Gains and Lowndes 2014). Within the construction industry gender rules around work exclude women because work is accepted as inappropriate (Galea 2018).

The exclusion and rejection of women from the construction workplace are informed by gendered biases that influence male workplace behaviours towards women. These behaviours result in "systemic gendered injustice" because of "male behaviours that self-interestedly structure the practices and cultures to make these workplaces unwelcoming to women" (Clarsen 2019, p. 35). Galea (2018) argues that masculine values and rules are reinforced and maintained through hegemonic consent and coercion. Hegemonic masculinity is the process of gendered power relations and order as informed by social practices. Hegemonic masculinity informs and shapes gendered rules. A gendered effect shapes how "gender actors work within the rules" (Galea 2018, p. 31), resulting in a lack of support for women in the construction industry. This is because social practices legitimise and give power to men (Hearn 2004). Men who "fit the hegemonic code or are complicit in maintaining it have practices and systems modelled around their experiences" (Galea 2018, p. 36). Therefore, they enjoy gifted unearned power through inherited systems of privilege, the product of membership to a social group based on gender. The dominant male group in the construction industry maintains existing hierarchical systems by granting benefits to those who are privileged by their gender (Bailey 1998).

In male-dominated environments, (Galea 2018), men "create, interpret, enforce, respect and communicate the rules as well as their perceptions of gender power relations and any prospect for change" (Galea 2018, p. 29). In other words, given their hierarchical power, men in an organisation shape, enact and influence "the rules or institutions associated with work, recruitment and advancement" (p. 29). Informal practices where men align with other men (homosociality) improve their access to power networks and perpetuate patterns of male dominance—adding to the advantage of men over women through their exclusion. (Galea 2018, p. 30) argues that the male-dominated "rules-in-us" in the construction industry "reinforces power asymmetry and gender patterns of advantage and disadvantage", including future policy direction and opportunities for reform (Chappell and Waylen 2013).

## 3. Gender Bias in the Workplace

The literature informing this research sits within a demand-side focus on gender segregation in the workplace. It focuses on the structural and organisational factors such as discrimination, employer preferences, and workplace culture as contributors to women's underrepresentation in many workplaces, including construction (Wright et al. 2015). Research on women working in construction has highlighted several issues that women face in the workplace due to their gender. These issues create barriers for women entering the workforce, maintaining employment, and advancing their careers. The demand-side barriers women face identified in existing research are exemplified by Rosa et al. (2017) and Turner et al. (2021), who identify sexist attitudes, behaviours and perceptions, long working hours, isolation on site, negative perceptions of women's capabilities, an expectation to mimic masculine behaviour, a lack of role models and mentors, family/work–life balance issues, slow/limited career progression, stress, being undervalued, and institutionalised discrimination. Similar issues emerge in other studies; for example, Navarro-Astor et al. (2017), in their review of the literature over 15 years from the year 2000, identified, in addition to women's career progression in the construction industry: gender stereotypes, allocation of posts and activities, working conditions, sexist culture, harassment and lack of respect, recruitment, lack of recognition, pay, and limited social networks.

While much work has been conducted to highlight issues occurring, less work has been undertaken to understand the gendered structural and cultural enablers for such issues, their cumulative and systemic effects, and how they can be effectively addressed. In other fields, underlying socio-structural drivers for gender inequality and associated harms have been researched more deeply. An excellent example of such research is Hutchison (2020) in their study of gendered bias facing women surgeons in Australia. Similar to women working in the Australian construction industry, women are a minority in the field of surgery, especially in leadership positions, and experience pay inequality. Hutchison (2020) cites this under-representation and disparity as due to a lack of work–life balance and parental leave, workload, and the unavailability of role models and mentors. As is the case in the construction sector, women surgeons also experience sexual discrimination and harassment. Hutchison (2020) identified implicit biases and epistemic injustice as playing significant roles in the prevalence of these workplace barriers leading to the under-representation of women in surgery, thus recommending the need for systemic approaches to meaningfully address the current workplace culture.

Of the four biases identified by Hutchison (2020), the first bias relates to workplace conditions, including the physical amenities, workplace climate, pay, and leave provisions. The second bias relates to credibility, where the knowledge and skills of women are challenged and judged as either in deficit or in excess compared with their male counterparts. Such credibility biases are informed by notions of epistemic injustice, whereby stereotypes lead to particular groups, such as women, being unable to contribute to the dominant discourses or practices they are a part of because they are not seen as legitimate contributors to knowledge (Fricker 2007). The third bias relates to the role and behavioural expectations of women held by colleagues, clients, and women. Finally, the fourth bias relates to objectification and expectations around women's appearance in the workplace.

The four gendered biases identified by Hutchison (2020) independently cause minor harm to women. However, these minor harms systemically interact, resulting in significant cumulative damage contributing to the under-representation of women surgeons, their well-being, career development, success, and retention. Hutchison (2020) concluded that only a systemic understanding of the interrelationship and effects of each bias experienced in the workplace would result in interventions to address the aggregate harms and challenge the support structures that enable and perpetuate them.

This paper draws on Hutchison's (2020) approach to explore the gender biases found in the Australian construction industry, the associated harms these have on women, and the resultant cumulative impacts. Such an approach is advantageous for describing what is happening and investigating how the underlying biases manifest to determine effective systemic interventions. Exploring if and how women experience these biases in trades and semi-skilled roles enables the identification of the individual harms and their cumulative impacts. Further, identifying the systemic relationship between these biases can inform opportunities to better understand interventions that could negate the complex socially constructed norms that promote hegemonic masculinity and its by-product, masculine privilege (Galea 2018).

## 4. Privilege and Epistemic Injustice

Membership in a social group defined by gender can generate male privilege or unearned advantage, and associated behaviours perpetuate bias and epistemic injustice. Epistemic injustice occurs when an individual is wronged in their capacity as a knower or communicator of knowledge (Fricker 2007). Epistemic injustice also occurs when a person's testimony is not given the proper recognition or credibility it deserves because of some prejudice or stereotype (testimonial injustice), or when a person's experiences or knowledge are not fully understood or articulated because the concepts or language needed to express them are not available or recognised in their culture or society (hermeneutical injustice) (Fricker 2007).

Privilege is usually invisible to those who benefit from it, given they are the norm. It enacts their legitimacy, inclusion, and authority within the social construction context. Consequently, discrimination against women often goes unnoticed or is denied. As Galea (2018) argues, "seeing" the discrimination would require the acknowledgment of the rules-in-use and the subsequent privilege and benefits resulting from the existing and maintained power imbalance. As individuals, men often feel powerless to acknowledge this injustice. Privileged people are less likely to recognise that others cannot access the same benefits, thus creating a culture of denial. Denying the power imbalance influences how the problem is understood and addressed, often resulting in its trivialisation, dismissal of existence, misrepresenting the situation, or in this instance, blaming women for the problems they face. Galea (2018) argues that this position of privilege results in an inability to "see" how the rules-in-use result in the justification of inequality in areas such as access to equal pay and benefits or a job. Further, the existing "meritocracy" (p. 42) between those with privilege and those without justifies and legitimises inequality within the social context or the workplace.

Fricker (2007) explains that epistemic injustice occurs because of identity prejudice that may arise in one of two ways: prejudice because of transactional injustice or structural injustice (Anderson 2012). To counter the inequity, the hearer must, according to Fricker (2007), check their prejudice by, as an individual, becoming more virtuous. Fricker (2007) argues that for a hearer to identify the impact of identity power in their credibility judgement, they must be able to identify "the impact on the speaker's social identity" (p. 91) and "the impact of their own social identity on their credibility judgement" (p. 91), which requires a corrective anti-prejudicial virtue awareness that is socially reflexive and critical. Fricker (2007) contends that correcting for prejudice in one's credibility judgement is when the hearer suspects prejudice in her credibility judgement and shifts:

> "intellectual gear out of spontaneous, unreflective mode and into active critical reflection to identify how far the suspected prejudice has influenced her judgement". (Fricker 2007, p. 91)

While there is a recognised need to correct prejudice, Fricker's (2007) arguments of virtue invoked through critical reflection have been challenged, given the operation of unconscious stereotypes. Avowed beliefs are so insulated from each other that individuals do not feel the cognitive dissonance that provokes "an occasion for critical reflection and the practice of virtue" (Anderson 2012, p. 168). Anderson (2012) contends that for virtue to become habitual, we need to know how to consciously practice, which can be problematic if we do not know where our prejudice lies and, as such, stresses the importance of structural solutions or virtue-based remedies for collective agents for epistemic justice. Such structural remedies should be designed to prevent cognitive biases from being triggered and to facilitate the conscious exercise of counteracting dispositions to fair assessment. Initiatives may include employment anti-discrimination targets and guidelines, which comprise institutional requirements based on explicit, objective measures rather than subjective assessments and accountability for discriminatory behaviours and decisions.

Understanding the cumulative effects of gender bias requires understanding the functioning system from which it is generated. As Galea (2018) argues, acts of privilege do not operate alone to maintain men's advantage and power status in construction. In the Australian construction industry, power, gendered actors, and gendered informal institutions work together to keep masculine privilege in place.

## 5. Methodology

The research methodology is guided by an interpretivist paradigm that seeks to construct an understanding of the experiences of women working in trades and semi-skilled roles in the construction industry and how their gender impacts their experience. Consistent with an interpretive methodology, knowledge is sought with an inherent understanding that truth or meaning comes into existence in and out of our engagement with the realities in our world (Denzin and Lincoln 2005). Subject and object emerge as partners in the



generation of meaning. From this perspective, understanding the social world can only be obtained from first-hand knowledge of the subject under investigation (Crotty 1989). This approach emphasises the analysis of the subjective accounts generated by 'getting inside' situations and involving oneself in everyday life.

Consequently, the research conducted and reported in this paper drew on semi-structured interviews with women in trades and semi-skilled roles. The interviews aimed to elicit the challenges associated with their experience on-site to explore the role of gender bias and its effects in identifying ways women could be meaningfully supported in the workplace. The interview focused on exploring their role/s, responsibilities, and day-to-day work experiences. All interviews were conducted over the phone.

Thematic analysis (Boyatzis 1998) was used to identify, analyse, and report patterns or themes within the dataset. Underpinning thematic analysis is the assumption that participants' thoughts and actions mirror their unique perspectives (Terry et al. 2017). Inductive thematic analysis, given the exploratory nature of the research, was deemed a suitable method from which an understanding of the complex and nuanced experiences of women working in the construction industry could emerge without the constraints of preconceived categories or theories.

The analysis followed the steps outlined by Nowell et al. (2017): familiarisation with the data, generating codes, and identifying, defining, and naming themes. To support the trustworthiness of the data, two researchers conducted the interviews, which were recorded and transcribed verbatim. In preparation for analysis, transcripts were de-identified, and each participant was allocated a unique identifier. Two researchers independently reviewed the transcripts, coded the data, and developed themes. Following this, the researchers met to compare themes and reach a consensus. Participant quotes were also used to evidence emergent themes when reporting the findings (Ballinger et al. 2004).

Trades and semi-skilled women working in construction in the Australian state of Victoria were recruited via an electronic or paper invitation distributed by the Victorian Construction, Forestry, Maritime, Mining and Energy Union (CFMMEU) and the Electrical Trades Union (ETU) to 450 members, including some members of the Plumbing and Pipe Trades Employees Union (PPTEU). Each member was followed up with a reminder email and an additional reminder text message in the case of the CFMMEU. Within the Australian commercial construction industry, unions play a multifaceted role including advocating for workers' rights in relation to fair pay, safe working conditions, and training. The CFMMEU, ETU, and PPTEU represent blue-collar workers engaged in trade work, labouring, machinery operation, and driving. In 2022, 355,300 people in the Australian State of Victoria worked in construction, with 38,700 (10.9%) holding union membership (Workplace Gender Equality Agency 2022). Historically, membership-based unions hold significant influence over the construction regulatory environment. Master Builders Victoria (an industry peak body) and Multiplex (a multi-national commercial construction company) also advertised the invitation to participate in this research project on their social media pages. Multiplex additionally placed a hardcopy poster on-site. The following 'women in construction' support groups posted the invitation to participate in this project on their website and/or Facebook page: Women in Trades Network, LadyTradies, SALT, Tradeswomen Australia, and TradeUP.

## 6. Results

### 6.1. Participant Demographics

Forty-three semi-structured interviews were conducted with women in trades or semi-skilled roles in construction. The interview duration ranged from 30 min to 2.5 h. At the time of the interview, participants were employed by a trade union (2%), head contractor (building and engineering companies) (19%), sub-contracting company (44%), and government organisation (12%), and 16% were sole traders or ran their own small business. In total, 84% of participants worked in the metropolitan area, and 16% worked in rural areas.

The age group of participants is summarised in Table 1, which ranged from 20–24 years to 60+ years.

**Table 1.** Age group of interview participants.

| Age | Participant Number |
|---|---|
| 20–24 years | 4 |
| 25–29 years | 6 |
| 30–34 years | 11 |
| 35–39 years | 5 |
| 40–44 years | 4 |
| 45–49 years | 6 |
| 50–54 years | 3 |
| 55–59 years | 2 |
| 60+ | 1 |

Note: One person did not give their age.

Participants worked in various sectors of the construction industry. Overall, 42% of women worked in commercial construction, 23% worked across commercial construction and infrastructure, 19% worked in infrastructure/civil works, 9% worked in rail, 5% worked in domestic, and 2% worked in the Technical and Further Education (TAFE) sector. In terms of years of experience in the industry, 36% of participants had worked in the construction industry for 1–4 years, 22% for 5–10 years, 14% for 11–15 years, 12% for 16–20 years, and 16% had worked in the industry for more the 21 years.

Participants' living arrangements and family structure comprised: 23% of interviewees lived alone, 19% lived with a partner, 18% lived with a partner and children, 11% lived with their parents and or sibling(s), 9% lived with a partner and were pregnant, 2% lived with their parents and their child, while a further 2% lived with their partner and parents.

*6.2. Themes*

Themes and subthemes emerging from the interviews are summarised in Table 2. Themes centred around challenges to credibility, the construction work environment, work role, objectification, and cumulative harm. The following sections expand on each of the themes.

**Table 2.** Themes and subthemes.

| Theme | Subthemes |
|---|---|
| Challenges to credibility | Misrecognition: Mistaken as a junior<br>Credibility excess<br>Credibility deficits |
| Workplace environment | Lack of workplace support<br>Amenities<br>Climate<br><ul><li>Hostile workplace.</li><li>Explicit sexism.</li><li>Harassment and assault.</li></ul>Working conditions<br><ul><li>Insecure work.</li><li>Lack of flexibility.</li></ul>Career development<br><ul><li>Lack of professional network.</li><li>Access to meaningful work.</li></ul> |
| Role | |
| Objectification | |
| Cumulative harm | |

*6.3. Challenges to Credibility*

From the interview data, there were many experiences women recalled where male colleagues questioned their credibility. As shown in Table 3, women's credibility was questioned regarding their intellectual and physical capabilities to perform their roles. The lack of credibility attributed to women was expressed in various ways, from male colleagues ignoring advice and input from women to questioning the legitimacy of their licence qualification or the legitimate experience they hold to perform competently in their role. Some women described their technical expertise as undervalued and subsequently were assigned roles like speaking with clients or administration when the secretary was absent. Another way bias in credibility judgements manifested was assuming that women on-site were in junior positions. While many comments from participants were about credibility deficits, some described excess; an example of this was described by one participant who had a client who insisted on paying her more for her work because she was a woman.

**Table 3.** Examples of challenges to women's credibility.

| **Misrecognition: Mistaken as a junior** |
|---|
| *"Every time I start a new job as an [stated trade], the first question that the guy I'm working with asks me is, "Are you an apprentice?""* |
| **Credibility excess** |
| *"He actually paid me extra money, and I'm, "No, I don't need it," and he's, "No, you girls are amazing. I'm really impressed…I'm not used to dealing with females working in this industry.""* |
| **Credibility deficits** |
| *"Because it (information) hasn't come from them [a man], it's not their idea, they haven't seen it, so, therefore, they can't see it that way."* <br> *"I have been looked over for higher roles, leadership roles because I'm a girl, and I've been openly told, 'You were the best candidate for this position, but boys won't listen to a girl. They won't listen to you."* <br> *"what Weetbix [breakfast cereal] box did you get your [trade] license out of?"* <br> *"women are perceived as she's not going to be strong enough to do this [the set task]."* <br> *"there is still this attitude that you're only here because you're a quota, or you're only here because you shagged the boss, or because you know someone."* <br> *"If the secretary is away, you'll be asked to go sit in the office and answer the phone for the day."* <br> *""All I want you to do today is talk to clients. Don't pick up a ladder, don't do anything." So, every time I worked with this guy, he would run around. A 65-year-old guy, he'd run around, do whatever he needed to do, and I would sit there and talk to clients."* <br> *"One guy I had, I was in a roof one day when I answered the phone, and he's like, "I need an electrician to come around. It's not a big job," I'm like, "Okay," and he goes, "Oh, but you're a girl," and I'm like, "Yeah?" He said, "It might involve getting under the house," and I'm like, "Well, I'm in a roof right now," and he's like, "But do you think you'd be able to do it?""* <br> *"I've walked up to someone I didn't know very well, and he's kind of a guru. He had more tickets and more qualifications than anybody in our depot. There was this tricky piece of equipment, so I walked up to a small group of them, and I said, 'What's this?' I was curious as to this new equipment that we were installing. His response was 'Secret men's business'."* |

Participants commented that femineity was not viewed as a positive attribute in the industry and created an implicit bias that women could not negotiate nor contribute to the workplace regarding knowledge and skills. The data indicated that this gendered bias resulted in women being attributed limited credibility. Furthermore, the interviewees' accounts indicated that many male colleagues held an implicit or explicit belief that women were not suited for construction work. As a result, women were perceived as a liability. The lack of credibility given to women led to them not being taken seriously on-site or listened to concerning practice.

Participants felt some employers viewed women as a burden rather than recognising the opportunity to develop and maintain inclusive workplaces and fair hiring practices. Participants noted being often misrecognised as either new to the industry regardless of age or not afforded recognition by men outside of the industry.

The interview data illustrates a distinction between roles and gender, with women not being considered worthy of access to an explanation of identified masculine work tasks and not being taken seriously concerning the work they have been employed to undertake. The perception that women do not belong on-site resulted in work tasks that failed to reflect their acquired training and ability. Some women felt they were employed because it was the politically correct thing to do. The gender-based credibility bias resulted in women often feeling not welcome or wanted, capable, or skilled.

### 6.4. Workplace Environment

Women described several factors associated with the workplace environment as unfairly impacting their role compared with their male colleagues. These included the level and type of support available, site amenities, workplace climate, working conditions and opportunities for career development.

The first workplace factor presented here relates to the provision of support. Participants reported not receiving the workplace support required for a successful work experience. As shown in the comments presented in Table 4, support in this context was understood as both the words and actions displayed by co-workers, management/supervisors, their employing organisations, and by support roles, including the Health and Safety Representative, Trade Union Shop Stewards, and their organisation's Human Resource group and policies.

**Table 4.** Examples of workplace factors that result in inequality: lack of workplace support.

| |
|---|
| *"It's not very often someone would stand up for me and be like don't talk to her like that."* |
| *"What I recall most is how disappointed I was in my mates that they wouldn't call it [inappropriate behaviour] out because they didn't want to be seen as weak".* |
| *"I was kicked off both jobs because of somebody else's behaviour and punished severely."* |
| *"When somebody uses a particular dirty - say the word 'c**t', lots of men are, and so often they would say, '[name] I cringe every time he f**king says it. Why don't you say something?'"* |
| *"The mentality was to figure it out or get out, and so you follow, and you learn, and if you can't do that, then you can't stay. It's that ruthless… people don't really care about teaching you the fundamentals of what you were doing."* |
| *"I would ask "Why are you doing that?" or "How do I do this?" You'd be right next to someone, and they would completely just pretend that they hadn't heard you."* |
| *"If I walk the job and say, 'Do you mind fixing this issue and this issue and this has to be done,' they'll [male co-workers] be like, 'Why? Who said that needs to be done?' And I'll say 'me,' and they'll take forever to do it [the task] because I've said it. If it's one of the other guys, they'll do it straight away. But because I'm asking them, they'll argue back and ask questions…basically, my role isn't taken seriously at all."* |
| *"Nobody wants to be the guy who stands up and says I agree with her; that's inappropriate because you're singling yourself out as a target, and you get put in that category of hysterical."* |

Women were frustrated and disappointed by the lack of support displayed by the male co-workers in their crew when poorly treated by other men on-site. This frustration was often conflated when male co-workers asked women to stand up to behaviour or language they found offensive. It was pointed out that the power that men have on-site over women precludes women from acting. It emphasised the importance of male co-workers calling out inappropriate language and behaviour, not just in the moment as support of solidarity but also in challenging the culture and standing up for the acceptance of women more generally across the industry.

When male co-workers witnessed unjust behaviour towards women, it was accepted by men as normal. Some women recognised that their male colleagues understood the power relationship and harm caused by such behaviours. However, the women believed their male colleagues felt powerless and fearful to openly discuss and challenge the power imbalance because the dominant culture would identify them as weak, different from the majority, and excluded from the privileged group.

The provision of amenities for women was noted as another workplace factor by participants. Table 5 presents participant comments regarding the provision of amenities

on site. Participants reported experiencing ongoing bias, both implicit and explicit, related to a workplace that was set up by men for men. Women cited that some employers viewed them as difficult to work with as they required separate amenities. Many women felt they were denied basic amenities and entitlements, specifically, separate women's toilets, sanitary bins, and work wear. By taking this position and denying women their rights, employers assume they should not have to create an inclusive workplace.

**Table 5.** Examples of workplace factors that result in inequality: amenities.

| |
|---|
| *"You still get the old school organisations that go 'we don't even have a female toilet.'"* |
| *"The single biggest issue in four years that I've ever faced on a construction-site is every single construction-site caters only to men…You never get a toilet."* |
| *"If a company cares about gender equality and all that sort of stuff with women, provide women in trades with clothing that fits."* |

An adverse workplace climate, including offensive behaviour from males on site, hostility, explicit sexism, harassment, and assault, were experienced by many of the women interviewed. More than half of the participants had experienced verbal harassment, offensive workplace banter, offensive personal comments about their appearance or capability, and highly confrontational, aggressive, and threatening communication. Table 6 illustrates examples of the workplace climate women experienced. Women described offensive workplace banter as dirty jokes, mild swearing, and sexist humour. Women also described humiliating and embarrassing comments directed at them in front of co-workers. Such behaviour was uncomfortable and difficult for women and undermined women's worth and ability. Women also experienced unwanted sexual advances, which left them feeling vulnerable. Many participants had experienced extreme threats of sexual violence by males trying to assert control over them to gain compliance. Inappropriate acts of harassment in the workplace included both verbal and alleged physical abuse ranging in severity and harm. Several participants reported being a victim of sexual violence.

**Table 6.** Examples of workplace factors that result in inequality: climate.

| **Hostile workplace** |
|---|
| *"They [men] always assume that at some point in time, you're going to turn into a b***ch."* |
| *"You can find yourself standing within an area that comes across as being quite aggressive. When there's stuff going on, some people can get quite uppity and aggressive, or come across aggressive."* |
| *"I expected, walking into a male dominated industry, to be treated differently as a female. I expected to be disrespected. I had all these expectations that they were met, and then probably surpassed."* |

| **Explicit sexism** |
|---|
| *"always overlooked… to the point now with my supervisor, I don't ask him what he wants me to do because I know he's going to give me some menial job. I'll go and find the work."* |
| *"I had guys hiding in scaffolds, watching me work. To them, it may seem harmless, but when there's one of you and a thousand of them, it probably just does put a bit of a crazy thought in the back of your mind."* |
| *"I've had colleagues who have openly said to me, "I've had two candidates. One said she was looking at starting a family, and the other wasn't, and I had to take on the male from a business perspective.""* |
| *"I had a guy who basically blackmailed me and said that he would [provide a negative performance review to the employer] because I [did not accept his advances]."* |

| **Harassment and assault** |
|---|
| *"A guy, the other week, goes ', Geez, you make those pants look good, and I was like ', What?'. I was like, oh my God, I don't know how to respond to this. That was wrong, he shouldn't have said that."* |
| *"[when asking a co-worker for assistance, he] started screaming at me all kinds of offensive things, along the lines of 'You're a woman, you have no right to tell me what to do, you're a piece of s**t', and he threatened to kill me. He threatened to rape me. He threatened to beat me to death."* |
| *"I've been flat out asked for sex on-site. I was shocked, and I was really uncomfortable."* |
| *""many people cross the line with threats and touching me… They pull out their genitals and tell you to touch it or suck it…"."* |
| *"locked in an on-site toilet…in the middle of summer for three hours, because they [male co-workers] thought it was funny."* |
| *"My first year in my apprenticeship, I was sexually assaulted."* |

Another workplace factor that impacted participants was workplace conditions, as highlighted in the examples presented in Table 7. Women identified that inflexible work hours prevented them from having children or returning to the workplace after having a baby; lack of access to maternity leave prevented them from spending time with their new-born infant; and loss of work. Some younger women planning to have a family expressed the need to identify alternative roles in the sector to enable them to continue working in their occupation. More than half of the women interviewed did not have children, and many of these participants stated that their work hours would not be achievable if they had children.

**Table 7.** Examples of workplace factors that result in inequality: working conditions.

| Insecure work |
| --- |
| *"I do shift work, which puts a big stress on the day-to-day running of my life."* |
| *"With contract work . . . you don't know how long it's going to last because you could only be contracted for six months to a year."* |
| *"It can definitely be a struggle not knowing if you don't have a job next week."* |
| *"men have stay-at-home partners."* |

| Lack of flexibility |
| --- |
| *"I guess if I had kids, I would not be able to do the work that I do now."* |
| *"Until I was actually pregnant, I thought I would make it work. It's definitely not going to happen like that."* |
| *"If you had a baby, that [the work hours] would be very, very difficult. . . You have to be on-site at 7 a.m. That is hard for a lot of people because a lot of kids don't start school till 8 o'clock or 9 o'clock."* |
| *"It's not as flexible in terms of the times you'd like to work, whether it's an afternoon shift or day shift. . .you've got to take whatever you can."* |
| *"There's no part-time work in construction. So, I started working for myself."* |

Some participants, pregnant at the time of their interview, identified a lack of proactive attempts to organise alternate work arrangements, given the risk their unborn children faced due to the nature of their work. In contrast for other participants, the nature of their employment contract influenced their ability to identify appropriate work and retain employment throughout their pregnancy. The tradeswoman on a permanent contract reported being transferred into a temporary administrative role. However, the other, a sole trader, was forced to stop work as she could not find more appropriate work. The organisation she was contracted to was not obligated to provide alternative opportunities. Pregnant women casually employed in semi-skilled roles reported a positive experience discussing alternative tasks with their foreman. In these instances, the work undertaken did not compromise the pregnancy. However, as a casual employee, post-pregnancy work was not guaranteed. Participants identified a lack of experience, understanding, and guidance from managers and supervisors to support pregnant women undertaking work activities that presented significant OHS risks to both mother and child.

The final workplace factor commonly noted by interview participants included the structures in place to support the career development of women. As shown in Table 8, participants identified that the insecure nature of their employment inhibited the development of their career pathways. Casual work and short to medium-term contracts were cited as problematic. Further, participants noted the lack of professional networks preventing them from gaining work. The opportunity for employment in leadership roles was cited as limited due to perceived capability, employers' perceptions, and cultural resistance to women in leadership roles. Participants reported being prevented from taking on leadership roles, despite having the appropriate experience and training. Promotion to a leadership role was often denied due to employers' perception of cultural barriers for women to successfully manage male workers. Some employers were aware of the cultural problems in the workplace. Still, rather than acting to address the issue, women were denied a promotion due to the exclusionary behaviour of male workers.

**Table 8.** Examples of workplace factors that result in inequality: career development.

| *Lack of professional network* |
|:---:|
| *"all word of mouth. That's how all the blokes get the work. . . know somebody who works at that company. It's not ever advertised openly anywhere."* |
| *"[male employees] all know each other. They all employ each other."* |
| *"like going to the pub on Friday and. . . there's also other things like guy's golf trips, which is the way they bond and mateship grows; therefore their position within the company also grows."* |

| *Access to (meaningful) work* |
|:---:|
| *"Nobody wants to hire a girl [apprentice]."* |
| *"I don't feel I'm respected at work at all. . .women can't run a shift. It has to be a boy. So, if it's my turn to step up, they will bring a boy in from another shift to cover me."* |
| *"Women are the first to be made redundant not because of your skill or your attributes that you bring, but because of your gender."* |

Participants also noted that promotion opportunities were discussed in informal social settings, which impacted women's ability to progress in an organisation. Women stated they were often excluded from social activities outside work hours.

*6.5. Role*

Many women interviewed commented that they felt they had to prove themselves worthy of their role in construction to their male counterparts. This indicated a clear gender stereotype about the role of women that had to be overcome in their work practice by demonstrating they had equal capability with men. The need to prove their worth and ability has clear links to participants' comments regarding credibility deficits and being listened to on-site as a legitimate practitioner and equal to that of male colleagues and supervisors. Examples of participants' comments regarding the need to prove themselves in the construction workplace are shown in Table 9.

**Table 9.** Examples of gender bias associated with the role.

| |
|:---:|
| *"I remember going on a lot of sites, and they either had the expectation that I'd be great—or had the expectation that I'd be s\*\*t. There was all this pre-expectation on me while no one gave a s\*\*t what that guy that I often worked with did."* |
| *"I earned that respect so that I could say something and people would listen, as opposed to me being dramatic, or me being a troublemaker."* |
| *"When a new person starts, although I've been with the company for seven years and it's their second week, I still feel like I have to prove myself to them."* |
| *"in construction, when you're the only female on the crew, your reputation then affects the next reputation of the woman that comes in after you."* |
| *"Very slowly and tactfully changing the way men perceive me and think of me or women in general. . . reprogramming the way men think".* |

Challenging gender stereotypes was seen as important, not just for the woman in question, but also for all women who followed them into those workplaces. Participants reported a sense of responsibility to set a positive example to make it easier for other women to enter the industry. These insights demonstrate that participants must challenge gender stereotypes to succeed in their roles. They believed that by doing so, they could make entry into the industry more accessible for other women. Consequently, the women interviewed felt the need to consistently prove themselves as highly skilled, committed, and capable. The pressure to perform was recognised as necessary given the perception, identified in the preceding section, of women not being able, capable, or suitable for the industry. Female workers in the construction industry reported working harder than their male co-workers because of the implicit bias that they did not belong. The desire to work harder was motivated by the need to prove they were capable and deserved to be accepted and allowed to complete the work they were employed to complete. Participants recognised

the critical relationship between respect and successful workplace experiences, and having to prove themselves was exhausting but required, regardless of their age and experience.

In addition to working harder and being better than their male co-workers at their jobs, over fifty per cent of the interviewed women adjusted their workplace behaviour to manage the challenges faced. Women emphasised that adapting their communication style was crucial to ensure their voices were heard. To counter gender-based assumptions, women were mindful of avoiding displays of weakness when experiencing gender based-challenges. Women used humour and sarcasm to counter unsettling or inappropriate behaviours. At times, they chose to ignore inappropriate behaviour from men in the belief that their response would signal that such behaviour was unacceptable.

Women highlighted the fundamental strategies they adopted to effectively navigate workplace challenges. Central to their approach was maintaining perspective and reframing negative experiences meant to test or undermine them, allowing them to concentrate on their career advancement. Engaging in activities that boost their mental and physical well-being helped them handle stress and enhance their capacity to tackle work-related tasks. Moreover, they actively dedicated themselves to developing skills and abilities to showcase their value. Recognising the significance of building support networks, they actively formed connections within and outside the workplace. These networks facilitated processing experiences with colleagues, family, and friends, enabling them to progress and manoeuvre their professional environments more effectively.

### 6.6. Objectification

As shown in Table 10, participants explicitly discussed feeling objectified due to gender and associated physical appearance in the workplace. Comments focused explicitly on their clothing at work. Participants reported being sexualised and objectified by their co-workers. This behaviour undermined women's abilities and resulted in their work and skills being judged on something other than merit. This behaviour was reported to have a corrosive effect on their self-esteem and confidence.

**Table 10.** Examples of objectification.

| |
|---|
| *"People would be nicer to me if I were wearing shorts. . .they would treat me differently depending on how tight my clothes were."* |
| *"I think if they're more attracted to a female, they're going to show them more respect and be a lot nicer. . . than to a girl that, who A, either rejects them or B, they don't find attractive. Then they're just going to be a complete a\*\*\*hole to them."* |
| *"Or just being male, the white privileged male who believe that they can still look at women as if they're a piece of meat."* |

### 6.7. Cumulative Harm

This study identified gender biases faced by women in the construction industry related to credibility, workplace environment factors, role, and objectification. Comments from participants illustrate how these biases are informed and manifest due to a fundamental assumption that women simply do not belong in the construction industry. The perception of construction as a male-gendered industry has normalised and legitimised systems of gendered rules shaped around masculine values, reflected in the following quote from one participant who explained some of the attitudes men held towards her and other female workers: *"Chicks are for cooking and cleaning. They're not supposed to be on-site. Get back in your own space"*, and *"You don't belong here. You're not going to last. You couldn't handle it. These things were being said to my face"*.

The assumption that the construction industry is for men perpetuates the exclusion faced by women because of the subsequent view that the women working in construction present a real threat to men, as explained by participants: *"I think they feel extremely threatened by us"* and *". . . he was saying how women are taking jobs of men"*.

Further, women threaten the existence of the masculine identity and ego as manifest in the current culture as they challenge the existing gendered norm and require their male colleagues to adjust their behaviours, as explained by a participant who commented: *"there's a woman on-site. We all have to change the way we behave"*.

## 7. Discussion

Inappropriate or abusive behaviour (not exhibited by all male co-workers but experienced by 63% of women interviewed) was informed and influenced by the masculine culture that dominates the construction industry and the associated attitudes and ideologies held by its workers, such as aggression and competitive behaviour (George and Loosemore 2019). Seventy-nine per cent of the women interviewed experienced an environment of unpleasant, inappropriate, and, on rare occasions, criminal workplace behaviour—generally involving bullying, discrimination, or sexual harassment. This finding is consistent with George and Loosemore (2019) who identified that the masculinity of construction workers in the Australian context relates to the physical and high-risk nature of the work that is reflected in the workplace culture. Male identity might additionally be drawn from sexuality, sometimes expressed through *"sexually explicit imagery, language and so-called humour on construction-sites"* (George and Loosemore 2019, p. 429), which is reflected in workplace behaviour and norms.

The perpetuation of male privilege and gender norms identified by this research influences the experience and opportunities of women in construction across their career lifecycles. The current workplace culture privileges men in the recruitment stage, reinforces privilege in retention, and culminates in the dominance of men in charge of construction projects and organisations. In turn, the increased chance of male promotion ensures the successful exclusion of women from the workplace across all levels and perpetuates the gender imbalance.

Workplace factors coupled with credibility challenges cumulatively impacted the career lifecycle of women in trades and semi-skilled roles. More than half of the women in this study reflected on their difficulties in gaining work in construction. Key obstacles included: a lack of access to professional networks needed to secure work often advertised informally, employers' negative perceptions of women, specifically that they perceived the work as too challenging for women, and a poor investment since women would not remain in the industry.

Women identified that a lack of employer acceptance presented additional challenges to their ability to gain employment and forge a successful career pathway. Similar to other Australian research on tradeswomen's career entry (e.g., Shewring 2009; Simon et al. 2016; Bridges et al. 2022), the women in our study reflected that their limited professional network put them at a disadvantage, as it limited their opportunity to identify employment opportunities and secure employment. This disadvantage was identified by women at all stages of their career: in gaining employment as part of their apprenticeship; upon completion of their apprenticeship; when seeking a career change; or when attempting to progress into more senior leadership positions. Participants acknowledged that access to jobs in the construction industry was primarily by "who", rather than "what", you know.

Participants highlighted a more profound underlying barrier to their career prospects: many construction men perceived women as incapable or "unfit" for the industry. A gendered work stereotype informed this perception held by men. The engrained masculine stereotype of construction work perpetuated the myth that women did not have the skills, physicality, or emotional ability to cope with the associated requirements of the job and the workplace. This result is consistent with the findings of Quay Connection (2014), Bridges et al. (2022), Clarsen (2019), and George and Loosemore (2019).

This lack of acceptance of women in the workplace challenged their ability to access work aligned with their skills. A central concern was an inability to access a progressive learning pathway with continuity, given their sometimes inconsistent work locations and supervisors due to working on multiple job sites for numerous organisations.

The lack of continuity of supervision, coupled with the gendered work stereotypes, resulted in inadequate workplace training. This finding is consistent with the research of (Bridges et al. 2022). The women interviewees cited that once they had completed their apprenticeship, regardless of their age, it was always assumed that upon commencing a job, they were an apprentice. Once their co-workers became aware that they were qualified, there was a firmly held scepticism about their capability to complete the work aligned to their qualifications. Furthermore, women in semi-skilled roles cited difficulty in moving out of entry-level roles such as traffic management, again due to the broad acceptance that this was a role most suitable for women working in construction, regardless of their qualifications. However, irrespective of their skill level, the barrier to progression into leadership positions persisted, as women stated that many male co-workers do not want to report to a woman. Due to their co-workers' attitudes, many participants were passed over for leadership positions despite their employer's open recognition of their capability.

Our findings identified that many women were denied appropriate physical amenities. In addition, women were often denied flexibility around work hours to accommodate parental responsibilities. The lack of provision for women devalued their workplace contribution and further constrained their career progression, which is in line with (Galea and Loosemore 2006). Given that women are the minority in the workplace and due to gendered cultural stereotypes that men do not take on caring roles, policies in this male-dominated industry have not been family friendly, leaving women with inadequate access to parental leave arrangements and flexible work hours. Maintaining employment is difficult as there is a perception that women will get married, fall pregnant, and never return. If they do return to work after having a baby, women are perceived as a potential liability given their caring responsibilities.

All four pregnant women in this study desired to return to work after having their children. However, they identified that workplace flexibility was not the norm. Because of the long hours, they explored alternative industry opportunities. Tradeswomen recognised that their training provided an opportunity to work for themselves, with family members also in construction, or in alternative workplaces outside of commercial or civil construction. Each of these options would provide them with more control over the hours they worked. Therefore, in order to return to work in the construction industry after having children, many tradeswomen were required to become sole traders or small business owners to enable a working arrangement that gave them school/childcare-friendly hours, flexibility, and family time.

One participant employed casually in a semi-skilled role in the civil sector was optimistic about returning to work after pregnancy, as she aimed to move into an office-based role on-site. This change in career path was enabled by well-established professional relationships with managers in her existing workplace and free training courses provided by her union, who were also critical supporters in this transition. However, the other woman working in a semi-skilled role without an alternate role to return to viewed returning to the industry as impossible due to the short notice for shifts and early starts, making it difficult to place her child into childcare. She also noted that her partner and extended family were also in the construction industry, meaning that the work demands of long and inflexible working hours would prohibit them from providing support.

Many participants believed that work–family conflict was not as problematic for their male co-workers as it was for them, which is consistent with previous research (Lingard and Francis 2009). Many participants described the potential impact of negative workplace experiences and the pressure of working long hours on their family life. Women identified the need to process the inappropriate behaviour experienced in their workplace before returning home, to ensure this did not negatively impact their home lives. According to participants, male co-workers did not generally recognise the additional pressure women experience when working full-time alongside their role as a parent. The women interviewed felt that there was not enough understanding from male co-workers and managers

and suggested that more education was required to create a more family-friendly and supportive atmosphere.

The results of this study exemplify a culture of epistemic injustice (Bailey 1998), where women in construction are denied recognition and credibility in both their presence and practice and when provided feedback about work practices. Further, the injustices they experience are perpetuated by a lack of understanding due to their minority status. Both forms of epistemic injustice have serious consequences, resulting in women being unfairly discredited or silenced, or important knowledge and experiences being ignored or overlooked. In addition to not being heard, women lack recognition as they are not perceived as equally valuable and deserving of respect. This lack of recognition from colleagues informs and erodes a lack of recognition of self as a unique and valuable individual. Further, the lack of recognition of women in the social structure and practices of the workplace results in institutional policies that fail to address their needs (Galea 2018; Bailey 1998). Understanding and addressing epistemic injustice is essential for promoting greater social justice and inclusivity in our societies and workplaces.

As outlined in the literature, homosociality describes the formation of close bonds among individuals of the same gender within social or professional settings (Galea et al. 2015). In the construction industry context, these norms significantly influence workplace culture, fostering and perpetuating gender-specific practices. This creates a gender imbalance, where men predominantly form work-related connections and friendships based on shared commonalities, thereby establishing a culture governed by gender-specific rules that inadvertently exclude women from the "norm".

Epistemic injustice occurs because the workplace "rules" and "norms" diminish the credibility of women, creating disadvantages (Fricker 2007). These rules and norms inform workplace policies and structures that do not accommodate the needs of a diverse workforce and result in issues such as unequal pay, lack of family friendly policies, and limited advancement opportunities for women. The privilege experienced by men as a result of homosociality prevents their ability to consciously or unconsciously recognise the gender bias experienced by women. This resultant workplace culture inhibits women from accessing the same opportunities as their male co-workers, such as workplace support, job opportunities, promotion, and acceptance.

Gendered peer support is crucial in responding to the harms that women experienced in this study. Representing and sharing experiences allows women to construct a self-narrative and share knowledge. Mutual recognition is vital for self-realisation and identity formation, which depends on developing self-confidence, self-respect, and self-esteem (Jackson 2018). These modes of relating practically to oneself can only be acquired and maintained intersubjectively by being recognised by others who also recognise them. As a result, the conditions for self-realisation depend on establishing relationships of mutual recognition and empathic responses. However, mutual recognition requires others who have shared the same harms to share their experiences to develop a balanced relationship between those harmed and the listener. A symmetrical relationship is essential as recognition reverses the roles of *"victim-testifier and hearer-respondent"* (Jackson 2018, p. 9). When a hearer responds with "me too", they become a reliable testifier and a victim, vulnerable to judgment. Therefore, mutual vulnerability empowers each other as moral and epistemic agents.

Given the unique nature of the experiences women face, connecting with other women in the same situation is essential. Women in construction should be aligned with mentors who are women and with support groups to share similar experiences. By supporting others through shared experiences, women can put into perspective the confronting nature of their own experiences. The ability to network with other women in the construction industry allows women to determine if their experiences are the "norm" and if their reactions are constructive in improving their circumstances. Therefore, connecting women in construction with other women both within and outside their workplace can enable

women who feel isolated to find support and be heard, and potentially prevent these women from exiting the industry as well as supporting their well-being.

In addition to communicating one's experience through testimony, hermeneutical resources, collectively shared conceptions, understandings, and imaginations that provide interpretations for social experiences, thus generating meaning, are also required. However, these interpretative resources are not always available due to the systemic prejudice regarding which social groups have the authority to interpret the world. The power and authority men have over women creates hermeneutical injustice given the gap in collective interpretive resources, placing women at an unfair disadvantage when it comes to making sense of their injustice (Fricker 2007). The social subordination experienced by women due to gender results in a different workplace experience and their inability to shape shared interpretive resources that would enable their recognition as epistemic subjects equal to men (Jackson 2018). Peer mentoring and support groups for women can help to address the existing hermeneutical injustice. Jackson (2018) argues that the process of intersubjective and mutual recognition is achieved when women can recognise shared experiences of injustice. In doing so, the hermeneutical injustice is made visible, the gap in hermeneutical resources is identified, and processes of mutual recognition are developed to address this gap. Honneth (1996) identifies that as ideas gain traction within a society, a subculture of interpretation is developed, and moral motives for a collective struggle for recognition emerge. Speaking out collectively raises the consciousness, brings *"submerged truths to the surface"* (Brownmiller 1999, p. 7), and the personal becomes the political (Jackson 2018).

Women in this study identified that their lack of value resulted in inappropriate behaviours and treatment and that they had no voice when seeking to report this harm. The pervasive silencing of women in construction resulted from "testimonial quieting", where a woman's testimony was silenced due to the hearer's failure to identify her as a knower. The lack of credibility given to women working in construction prevents them from speaking out against the unjust behaviours they experience, denying them status as a "genuine victim" (Jackson 2018). When men respond to a woman's testimony in disbelief, they deny women the status of someone who can authoritatively speak to the facts of the event they experienced. Further, they deny her the status of someone others can rely upon to gain knowledge about the world.

The consequence of this testimonial injustice is that it alienates a woman from her self-relation and her relation to others, and her diminished status erodes her self-confidence to effect change. She is less likely to improve her working conditions. Women come to accept the judgement that they are at fault and do not belong in the industry, that their gender and actions resulted in the experiences, and that they deserved to suffer them. The individual harms experienced by our participants collectively contribute to their compromised well-being, career development, and retention in the industry. In addition, the realisation that their testimony of harm or victimisation has little chance of gaining uptake coerces women into silence because speaking of their harms would risk future harm to themselves (Jackson 2018).

As identified in this research, male workers often respond with hostility instead of providing sympathy and social support, and thereby women become socially isolated and ostracised. Denying the primary wrong and harms women experience working in the industry results in denial of *"access to the emotional, social, medial and legal responses that status warrants"* (Jackson 2018, p. 7). Women are left to cope with the harm they endure on their own and locate support services outside their workplace.

Women felt that their workplaces did not provide them with consequence-free communication pathways that allowed them to "speak up" when experiencing inappropriate behaviour or to report illegal behaviour they had been subjected to, without fear of judgment or punishment, especially by those in senior or leadership positions. Women said this resulted from the complexity of on-site reporting structures due to labour procurement contracts coupled with the dominant masculine culture and gendered work stereotypes. A complex workplace is characterised by multiple layers of supervision and manage-

ment reporting lines manifested differently in each workplace. The multiple layers of management, while presenting multiple opportunities for potential support for women, resulted in significant challenges, given the widely varying levels of acceptance of women in the industry.

Galea (2018) argues that given privilege is invisible to those in a position of power when it is denied or challenged, and the disruption of the status quo or position of power results in their discomfort, defence, resistance and, in some cases, backlash. Backlash is *"the response to a perceived disruption to the power held within the existing social order and gender power relations"* (Galea 2018, pp. 44–45).

Without clear and transparent processes and consequences, women felt fearful of reporting the unacceptable behaviour they had experienced. Compounding their fear of reporting was a fear of retribution. Forty per cent of interview participants believed they would be punished if inappropriate behaviour was reported. The likely punishments they feared were:

1. Being labelled as 'difficult';
2. Placing a strain on working relationships;
3. Being stood down from work;
4. The woman's sub-contractor organisation potentially losing future contracts;
5. Women being removed from their workplace and the perpetrator of illegal, abusive, or discriminatory behaviour going unpunished.

Our findings suggest that many male co-workers failed to speak up when witnessing inappropriate behaviour. The lack of intervention by male co-workers can be attributed to several factors, including the bystander effect, apathy, social norms that women are not valued or accepted, and the ambiguity associated with behaviour aligned with a masculine culture. Our study identified that women expressed a need for workplace support when they had been overwhelmed by rude, disrespectful, or aggressive behaviour.

An environment that recognises the status of every member as a knower and potential contributor or one that is epistemically inclusive and reflective requires institutional protection of epistemic friction. Epistemic friction is the coexistence of significantly different perspectives. Kwok (2021) argues that epistemic friction creates a safe environment and encourages individuals to be self-critical, compare their beliefs, and recognise cognitive gaps. Epistemic friction requires institutional protection and facilitation, especially in a hierarchical workplace such as construction, where men hold power and women do not. Speaking out is essential as it communicates women's experiences required to create new hermeneutical resources (Kwok 2021). However, articulating epistemic friction requires institutional protective measures in the form of organisational policies; otherwise, women in construction are afraid to speak out publicly because of fear of punishment.

Participants raised the issue of a lack of organisational governance and the absence of policies and procedures to address the harm women experience. Without support from managers and head contractors, workers who behave inappropriately towards women will continue to perpetuate a culture of epistemic injustice.

## 8. Conclusions

This paper presented data from a study involving interviews with 43 trades and semi-skilled workers in the Australian construction industry. This study identified a range of gendered biases that affect women's experience on-site and that, combined, cause cumulative harms and disadvantages for women. As identified in the results, these harms lead to isolation, exclusion, and inability to access employment or work that aligns with their training and experience and prevent women from developing their abilities and presence in the industry. The male-informed culture of construction creates epistemic injustice and a lack of recognition due to gender bias, leading to harm to women. These harms have a cumulative impact, resulting in systemic and structural discrimination. Recognising these cumulative harms is essential in enhancing women's experiences and

directly influencing the attraction, retention, and well-being of women in trades and semi-skilled roles.

Addressing homosocial norms in the construction sector necessitates a concerted effort to create a more inclusive environment that embraces diversity and encourages the active participation of women. Promoting diversity, implementing inclusive policies, providing equal opportunities, and challenging stereotypical norms can help to create a more welcoming and equitable workplace for everyone. This multifaceted approach requires the challenging of norms, reforming workplace culture, ensuring equal opportunities, and providing robust support and mentorship for women pursuing careers in construction. Initiatives aimed at fostering an inclusive and supportive environment can significantly contribute to promoting diversity and equity within the industry.

Importantly, this study contributes to an understanding that discrimination against women in construction is not a matter of individuals but a matter of culture and structures in the industry. A systemic response that targets structural enablers of gendered biases by men towards women is needed to address the epistemic injustices and other harms experienced by women. Along with this, structural changes to policy and regulations must occur that include employment anti-discrimination targets and guidelines and explicit objective measures. The Victorian Government in Australia is currently undertaking such work using a suite of initiatives with gendered targets and requirements for organisational reporting (including for sub-contractors) on large government contracts against gender equity target actions to improve women's inclusion, support, and experiences on-site. Such policy initiatives work towards structural changes in the industry to help address many of the biases described by the research participants. Further research is needed, however, on the effects of the Victorian Government policy in addressing the issues identified in this paper.

This research identified that women's support networks are vital in providing women with recognition, safety, and opportunities in the workplace. This is especially important when support from men in their workplace is unavailable. Of concern, our findings highlight that some men found the inappropriate behaviour of their male colleagues abhorrent, yet they rarely spoke up. Examining the lack of allyship amongst men who witness inappropriate or unfair behaviours or cultural norms is recommended as an important area of future research that can inform interventions designed to educate and enable industry change.

**Author Contributions:** Conceptualization, S.H., M.T. and O.S; methodology, S.H. and M.T.; formal analysis, S.H.; investigation, S.H.; resources, S.H., M.T. and O.S.; writing—original draft preparation, S.H., M.T. and O.S.; writing—review and editing; project administration, S.H.; funding acquisition, S.H. All authors have read and agreed to the published version of the manuscript.

**Funding:** This research was funded by the Victorian Government in Australia as part of the Victorian Women in Construction Strategy 2019–2022 Building Gender Equality program.

**Institutional Review Board Statement:** The study was conducted in accordance with the Declaration of Helsinki and approved by Human Research Ethics Committee of RMIT University (22934 13 May 2020).

**Informed Consent Statement:** Informed consent was obtained from all subjects involved in the study.

**Data Availability Statement:** The data presented in this study are not publicly available due to privacy and ethical restrictions.

**Conflicts of Interest:** The authors declare no conflict of interest.

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
