# Peer review of "Gender Bias in the Australian Construction Industry: Women’s Experience in Trades and Semi-Skilled Roles"

_socsci, doi:10.3390/socsci12110627_

Round 1
Reviewer 1 Report
Comments and Suggestions for Authors
This paper is particularly welcome as the lens and focus for this research addresses the conspicuous lack of analysis within the construction discipline that delineates between experiences of trade/semi-skilled roles and professional roles. This narrowed focus for exploring gender parity within the Australian construction industry has therefore identified issues and recommendations which have specificity of action, ownership, and accountability.
Despite the narrowed frame for research, a good number of participants have been involved in the qualitative investigation, providing confidence in the reliability and generalisability of results.
The literature review provides a good synopsis of the key issues, challenges, and authors relating to the topic and does so succinctly and with relevance to the research objectives. In doing so, the literature review provides not only a pragmatic review of the current and recent context of women in construction, but also of the philosophical underpinnings that describe how we can understand the context in social and cultural terms.
This consideration of the philosophical underpinnings drives a methodology that is consequently well conceived and described. However, how the authors got from transcript to theme is not explained. It would be useful for the reader to understand how the themes were generated e.g. completely data driven, or testing initially hypothesised themes? What was the method used here?
Results and thematic observations are well supported by evidence via quotes from the interview transcripts which are at times shocking but vital to highlight the lived experiences of women in construction.
The discussion section accurately reflects the results and summarises the key challenges and priorities for change. This section could be improved further by expanding a little on the drivers and/or factors that influence the status quo e.g. the increased chance of men getting promoted (amongst other male privileges) driven by norms such as homosociality which is well documented in the gender debate within the construction disciplines.
The paper shares some important themes that require further research, especially those in areas that remain unexplored within the discipline literature to date, such as the lack of allyship amongst men who witness inappropriate or unfair behaviours or cultural norms. These should be explicitly identified to expand the recommendations for future research within the conclusions section.
Comments on the Quality of English Language
A couple of inaccuracies/typos were noted:
At 6.1, 43 interviews were recorded, but the abstract and on p2 (line 69) indicate 44.
In the quotes on page 8 - should it be Weetabix?
Also on page8- typo - femininity? At line 324
Author Response
Thank you for your feedback.
We have revised and responded to your comments.
Please see the attached document for our detailed responses.

Reviewer 2 Report
Comments and Suggestions for Authors
This paper is almost ready for publication. I only present some minor observations, which could benefit the authors in finalising the paper.
It seems clear, based on the evidence presented in the paper, that there is question about systemic and structural bias in women´s position in the Australian construction industry. It could be stated clearer in the conclusions, that the discrimination against women is basically not a matter of individuals, but a matter of culture and structures in the industry.
In rows 29-38 the rapid decline of women in construction is described, but wothout adequate explanations for the development. This raises the question why has this happened?
Row 147 onwards: the position of women surgeons is discussed, but this part is not properly bound with women in construction industry. Consider what to do with this.
The invitation to interviews was distributed via unions. It is not explained in the paper, what the role of unions is in the Australian constructions industry, so the reader does not know whether they are representative to all workers or not or whether certain groups of workers are largely outside of unions.
Author Response

(The authors gave the same response as above.)

Reviewer 3 Report
Comments and Suggestions for Authors
The topic is interesting and I'm glad to have opportunity to review this paper.
There are some flaws need to be considered:
-The number of participants in the study is not mentioned. A small sample size can limit the generalizability of the findings. It's important for academic research to have a sufficiently large and diverse sample to draw meaningful conclusions.
-The paper doesn't provide details about the data collection methods and sources. It's important for academic papers to be transparent about the research methodology, including how data was gathered and analyzed.
- The paper should acknowledge any potential researcher bias or subjectivity in data collection and analysis. Additionally, the role of personal values or perspectives in shaping the research should be considered.
: While the paper mentions that challenging gender stereotypes is seen as important, it lacks a deeper discussion of how these women attempted to challenge these stereotypes or the strategies they employed. A more comprehensive exploration of this aspect would provide valuable insights.
- In the discussion section, interpret the research findings, connecting them to existing literature. Discuss the implications of the identified gender biases for women in the construction industry and offer potential solutions or recommendations.
- While the paper identifies the need for structural changes in policies and regulations to address the gender biases and injustices, it doesn't offer specific recommendations or propose policy changes. It's essential to not only identify issues but also suggest potential solutions.
- Provide clear recommendations for addressing the gender biases and injustices identified in the paper. This could include suggestions for policy changes, workplace interventions, or industry practices that can promote gender equity.
Comments on the Quality of English LanguageIt's ok.
Author Response

(The authors gave the same response as above.)

Round 2
Reviewer 3 Report
Comments and Suggestions for Authors
Authors have done good job to improve the paper.